# The Effect of Wharton Jelly-Derived Mesenchymal Stromal Cells and Their Conditioned Media in the Treatment of a Rat Spinal Cord Injury

**DOI:** 10.3390/ijms20184516

**Published:** 2019-09-12

**Authors:** Milada Chudickova, Irena Vackova, Lucia Machova Urdzikova, Pavlina Jancova, Kristyna Kekulova, Monika Rehorova, Karolina Turnovcova, Pavla Jendelova, Sarka Kubinova

**Affiliations:** 1Institute of Experimental Medicine of the Czech Academy of Sciences, Videnska 1083, 14200 Prague, Czech Republic; 2Charles University, Second Faculty of Medicine, 14220 Prague, Czech Republic

**Keywords:** spinal cord injury, mesenchymal stem cells, Wharton’s jelly, conditioned medium, cell secretome, cell therapy

## Abstract

The transplantation of Wharton’s jelly derived mesenchymal stromal cells (WJ-MSCs) possesses therapeutic potential for the treatment of a spinal cord injury (SCI). Generally, the main effect of MSCs is mediated by their paracrine potential. Therefore, application of WJ-MSC derived conditioned media (CM) is an acknowledged approach for how to bypass the limited survival of transplanted cells. In this study, we compared the effect of human WJ-MSCs and their CM in the treatment of SCI in rats. WJ-MSCs and their CM were intrathecally transplanted in the three consecutive weeks following the induction of a balloon compression lesion. Behavioral analyses were carried out up to 9 weeks after the SCI and revealed significant improvement after the treatment with WJ-MSCs and CM, compared to the saline control. Both WJ-MSCs and CM treatment resulted in a higher amount of spared gray and white matter and enhanced expression of genes related to axonal growth. However, only the CM treatment further improved axonal sprouting and reduced the number of reactive astrocytes in the lesion area. On the other hand, WJ-MSCs enhanced the expression of inflammatory and chemotactic markers in plasma, which indicates a systemic immunological response to xenogeneic cell transplantation. Our results confirmed that WJ-MSC derived CM offer an alternative to direct stem cell transplantation for the treatment of SCI.

## 1. Introduction

Spinal cord injury (SCI) represents a severe trauma for which an effective treatment is still not available. The limited neuroregeneration capability of the adult central nervous system (CNS) often results in a permanent loss of the motor and sensory functions below the injury, followed by a corresponding dysfunction and serious socio-economic consequences. In preclinical studies, a broad spectrum of strategies have been used in order to reduce oxidative stress, prevent apoptosis, promote the function of spared axons, induce new axon growth or replace lost cells using stem cell transplantation [1,2,3].

In this regard, stem cell therapy and in particular, mesenchymal stromal cell (MSCs) transplantation, has been proposed as an effective, safe and feasible alternative method for SCI repair [4,5]. The great potential of MSCs for cell therapy has been associated with the possibility of their isolation from various tissues within the body including bone marrow (BM), adipose tissue (AT), dental pulp, placenta, umbilical cord blood or Wharton’s jelly (WJ) without any ethical constrains, along with easy culture and a high proliferative rate.

The beneficial effect of MSC transplantation in SCI has been demonstrated in numerous studies after intravenous [6], intrathecal [7,8] or intraparenchymal [9] applications, suggesting that the regenerative effects promoted by MSCs are mainly associated with their paracrine effect [10]. MSCs certainly secrete a broad spectrum of factors, such as growth factors, cytokines, chemokines and immunomodulatory molecules that may stimulate neurogenesis and angiogenesis, or inhibit pro-inflammatory responses, apoptosis and glial scaring [11,12]. The released molecules can be present in a soluble form or internalized in the extracellular vesicles, namely exosomes and microvesicles [13].

The paracrine mechanisms mediated by the factors released from MSCs are generally considered to play a major role in improvement of the disease progression. Correspondingly, there are an increasing number of studies that demonstrate that the therapeutic effect of MCSs can be mediated by their secretome, which can be present in the medium where the stem cells were cultured [14,15].

Transplantation of MSC secretomes, also called conditioned medium (CM), or purified MSC-derived extracellular vesicles, have shown similar therapeutic effects and functional improvement as direct stem cell transplantation in various disease models [13]. Targeting CNS, the MSC secretome has been shown to promote the dopaminergic neuronal cell survival and motor function in an animal model of Parkinson’s disease [16], has promoted repair and improved recovery in preclinical models of cerebral ischemia [17] and attenuated the degeneration of axons and myelin of spinal motor neurons in a model of spinocerebellar ataxia [18]. The anti-inflammatory effect of MSC secretome has also been demonstrated in experimental models of multiple sclerosis [19] and Alzheimer’s disease [20].

In SCI repair, there are an increasing number of reports demonstrating the effects of MSC-derived CM or its derivatives after both local and systemic administration. Repeated intrathecal application of CM derived from rat BM-MSCs has had effects on locomotor improvements and tissue repair, including axonal regeneration, and attenuated inflammation in rat SCI [21,22]. Similarly, intrathecal administration of BM-MSCs derived CM improved functional recovery, reduced cystic cavity and exhibited a pro-angiogenic effect; however it had no effect on astrogliosis, macrophage invasion and axonal regrowth [23].

After intravenous administration, CM derived from BM-MSC significantly improved behavioral recovery of rat SCI, increased the densities of axons in the lesion site and increased autophage-related proteins in the injured spinal cords [24]. Furthermore, intravenous administration of MSC-derived exosomes or extracellular vesicles revealed pro-regenerative effects including functional recovery, enhanced axonal regeneration, pro-angiogenic properties, attenuation of neuronal cell apoptosis and lesion size and suppression of glial scar formation and inflammation [25,26,27].

In addition to MSCs, functional recovery, anti-inflammatory and regenerative effects after SCI have been demonstrated after systemic treatment with peripheral blood mononuclear cells-derived secretome [28], conditioned medium derived from olfactory ensheathing cells [29] and neural stem cells [30].

Systematic studies have confirmed that the paracrine activity of MSCs from different tissue sources is dependent upon origin. However, the source of MSCs suitable for cellular therapy and the optimal microenvironment to improve their neurotrophic/neuroprotective activity is still not fully determined.

Along with others, we have shown that MSCs derived from BM, AT or WJ differ in their secretion of neurotrophic, neuroprotective or antioxidative factors [31], however, all of these cell populations shared the capability of secreting important neuroregulatory molecules [12].

We have previously demonstrated that intrathecal transplantation of BM-MSCs and WJ-MSCs had a positive impact on neural tissue regeneration in SCI models, and that this effect increased with the dose [7,8]. Furthermore, MSCs can be seeded in appropriate scaffolds that create structural support for the transplanted cells and endogenous tissue infiltration [32,33].

Among the MSC sources, WJ-MSCs have been proved as advantageous in their high proliferative potential with low senescence up to the high passage, while their neuroprotective and angiogenic effects have been found to be superior compared to bone marrow MSCs [34]. In addition to the best proliferative rate, others and we have found higher production of neurotrophic factors by WJ-MSCs, compared to BM-MSCs or AT-MSCs, respectively [31,35].

Based on our previous results from WJ-MSCs transplantation in SCI repair [8], we aimed to investigate in this study whether CM produced by WJ-MSCs can be used as a sufficient therapeutic tool, with the ability to fully replace direct cell delivery.

We investigated the protein composition of WJ-MSC derived CM and its neurotrophic effect on axonal growth of dorsal root ganglia (DRG) neurons in vitro. In the in vivo model of SCI, we compared the effect of WJ-MSCs and their CM after intrathecal application in the 1st, 2nd and 3rd weeks following the induction of balloon compression lesions. The functional outcome was assessed during 9 weeks by sensory-motor tests (Basso, Beattie and Bresnahan test—BBB, beam walk test, plantar test). Furthermore, histological and immunohistochemical analyses were performed to evaluate the gray and white matter sparing, axonal sprouting and glial scaring. In addition, relative gene expression of selected markers related to neural regeneration and inflammation, as well as proteomic analyses of plasma and cerebrospinal fluid were performed (Figure 1).

Our results have proven that CM derived from human WJ-MSCs improved repair after SCI in rats and that this effect is comparable or even superior to direct human cell transplantation.

## 2. Results

### 2.1. Proteomic Analysis of WJ-MSC Derived CM

WJ-MSCs were characterized in the 3rd passage and fulfilled the ISCT criteria (The International Society for Cellular Therapy). They were plastic adherent, positive for CD105, CD73 and CD90 and lacked expression of CD45, CD34, CD14 and HLA-DR surface molecules; and differentiated to osteoblasts, adipocytes and chondroblasts [8].

Using proteomic analysis of WJ-MSC-derived CM, we detected the highest amount of the hepatocyte growth factor (HGF) and in a lesser amount other neurotrophic factors, such as brain-derived neurotrophic factor (BDNF), basic nerve growth factor (bNGF), fibroblast growth factor-2 (FGF2) and other factors related to migration—monocyte chemotactic protein 1 (MCP1), inflammation—interleukin 6 (IL6), MSC mobilization and tissue regeneration—stromal cell-derived factor-1 alpha (SDF1α), angiogenesis—vascular endothelial growth factor A (VEGFA) and immunosuppression—indoleamine 2,3-dioxygenase (IDO), soluble intercellular adhesion molecule-1 (sICAM1) and soluble vascular cell adhesion molecule-1 (sVCAM1; Figure 2). None of these markers were observed in control non-conditioned (cM) medium. The neurotrophic properties of CM were then confirmed by the DRG axon growth, which was significantly enhanced compared to cM medium (Figure A1).

### 2.2. Behavioral Testing

#### 2.2.1. Beam Walk Test

The beam walk (BW) test was performed from the 3rd week after the injury. Beam walk time (Figure 3A) shows the time the animal needs to cross the 1 m long flat beam. Beam walk score (Figure 3B) refers to advanced locomotive skills, maintaining of balance and motor coordination of the animal. The BW time was significantly shorter in all treated groups, cM, CM and WJ-MSCs, when compared to the saline treated control group from the 4th week after the injury and gradually decreased up to the end of the experiment. Moreover, the CM group revealed significantly shorter BW time in the 6th week when compared to the WJ-MSCs (*p* < 0.01), and in the 7th week when compared to WJ-MSCs (*p* < 0.001) and cM (*p* < 0.01), respectively. Similarly, the BW score was significantly higher (*p* < 0.001) in all treated groups when compared to the saline treated control from the 3rd week and gradually increased up to the end of the experiment, while the control group remained unimproved. The CM group then reached a significantly higher BW score than the WJ-MSCs in the 3rd (*p* < 0.05), 7th (*p* < 0.001) and 8th (*p* < 0.01) weeks after lesion induction.

#### 2.2.2. BBB Test

Recovery of the hind limb locomotor function was examined every week after the SCI using the BBB motor performance test (Figure 3C). The 1st week after surgery, the control (saline treated) rats were paraplegic and scored 0–1 on the BBB scale. Treatment with WJ-MSCs and cM significantly increased the score up to 2, while the best results were reached by CM group, which scored up to 4 and was significantly higher (*p* < 0.05) than all other groups. A rapid improvement was observed up to the 3rd week after the injury, followed by slower and gradual improvement in all groups up to the end of the experiment. From the 2nd week, all treated groups recovered significantly better than the control group (*p* < 0.001), but without significance between each other.

#### 2.2.3. Plantar Test

To assess thermal nociception, stimulation by the plantar test (Ugo Basile) was used (Figure 3D). In the week before SCI, the animals were tested three times, and the results showed no statistically significant differences among the groups. From the 1st week after injury, the control (saline) group showed significantly decreased (*p* < 0.001) latency in comparison to all three treated groups. In the 1st week, the latency was significantly higher in the WJ-MSCs treated group, compared to both the CM (*p* < 0.001) and cM (*p* < 0.01) group, respectively. In the 3rd week, the CM treated group showed a significant decrease of withdrawal latency, compared to both the cM (*p* < 0.05) and WJ-MSCs (*p* < 0.01) group, respectively. From the 4th week up to the end of the experiment, the latency of withdrawal in the treated groups returned to the levels before the injury, typical for a healthy animal, without significant difference between groups.

### 2.3. Histological Analyses

#### 2.3.1. White and Gray Matter Sparing

The treatment effect on the sparing of white and gray matter was evaluated 9 weeks after the SCI using cresyl-violet luxol fast blue stained serial cross sections (Figure 4). The significant tissue sparing was found in the CM and WJ-MSCs group in both the cranial and caudal parts of the lesion in the white matter (Figure 4A), and in the cranial part of the lesion in the gray matter when compared to saline treated control animals (Figure 4B). The cM treatment revealed a tendency of improvement in tissue sparing similarly as WJ-MSCs and CM, however it was statistically significant only in the cranial part of the white matter.

#### 2.3.2. Glial Scaring

Analysis of glial fibrillary acidic protein (GFAP) staining did not reveal a considerable reduction in the size of the glial scar after the treatment, apart from the cranial part of the lesion, where we detected a significant reduction of glial scar size after CM treatment (*p* < 0.01; 5 mm cranially from the centre of lesion) and WJ-MSC treatment (*p* < 0.05; 4 mm cranially from the centre of lesion; Figure 5A). The number of reactive protoplasmatic astrocytes was significantly reduced after CM and cM treatment in comparison to the control (*p* < 0.05), while the WJ-MSCs group did not reveal any significant changes throughout the whole lesion (Figure 5B). Moreover, the number of reactive astrocytes in CM group was significantly lower compared to WJ-MSCs group. The representative images of GFAP staining of astrocytes are then shown in Figure 6.

#### 2.3.3. Axonal Sprouting

Analysis of GAP43 staining was used to reveal newly sprouted axonal fibers. The increase in the number of GAP43 positive fibers was found in all three treated groups (Figure 7A). A significantly higher number of GAP 43 positive fibers was found in the WJ-MSCs treated group (7x ×, compared to the saline treated control, *p* < 0.01). The highest number of GAP43 fibers was found in CM treated group (13x×, compared to the saline treated control, *p* < 0.001). The representative images of GAP43 staining of newly sprouted axonal fibers are shown in Figure 7B.

### 2.4. Gene Expression Analysis

The following genes were selected for multi-sided screening of different cellular functions: Chemokines (*Ccl3* and *Ccl5*), genes associated with immunomodulation and inflammation (*Tnfα*, *Il1b*, *Mrc1*, *Irf5* and *Nfκb*) and genes related to neuroregeneration (*Fgf2*, *Gap43* and *Gfap*). Changes in the mRNA expression were determined 4 and 9 weeks after the injury (four animals from each group).

After 4 weeks, qPCR analysis revealed a significant up-regulation in relative expression of genes coding chemokines *Ccl5* (RANTES), proinflammatory cytokines *Il1b*, and M1 proinflammatory macrophages (*Mrc1*) in the WJ-MSCs treated group, when compared to the saline treated control (Figure 8A, B). In the case of *Ccl5*, *Tnfa* and *Mrc1*, the increase of expression in the WJ-MSCS group was statistically significant in comparison to the CM treated group. On the contrary, CM treatment resulted in a significant decrease in the expression of both M1 (*Irf5*) and M2 (*Mrc1*) macrophage markers 4 weeks after the injury.

After 9 weeks (Figure 8C), the gene expression of *Il1b* and *Tnfa* was no longer detectable (Δ*C*t 40) and levels of *Ccl3* and *Ccl5* were very close to the detection limit (Δ*C*t 38–39), thus excluded from further analysis.

A significant up-regulation was found for *Mrc1* and *Nfκb* genes in all three treated groups. Expression of *Gap43* and *Fgf2* in CM and WJ-MSCs groups was then significantly higher than the saline and cM treated control.

In the case of *Gfap* expression, no significant changes were detected after 4 weeks, while after 9 weeks, a significant down-regulation was found in the CM and cM group, but not in the WJ-MSCs group when compared to the saline treated control. Moreover, the down-regulation of GFAP in CM group was statistically significant, when compared to both the WJ-MSCs and cM group, respectively.

Relative expression of *Nfκb* was significantly increased in all three treated groups, compared to the saline treated control. Additionally, both CM and WJ-MSCs groups were significantly increased in comparison to the cM group.

### 2.5. Cytokine Profile in Blood Serum and CSF

The level of IL-1β, IL-2, IL-4, IL-6, IL-10, IL-12p70, IFN-γ, MIP-1α (CCL3) and RANTES (CCL5) was analyzed 4 weeks after SCI in blood serum and in the cerebrospinal fluid (CSF; Figure 9). From all these markers, only RANTES was found above the detection limit in both the serum and CSF, and TNF-α, MIP-1α, IL-1β and IL-2 were detected in the serum. Of note, all these detected markers were significantly increased after the treatment with WJ-MSCs, which suggests a systemic immunological response to the xenogeneic cell transplantation. In CSF, the RANTES level was significantly higher after WJ-MSCs treatment than in the healthy control (*p* < 0.05) and after CM treatment (*p* < 0.05). In the serum, the RANTES level significantly increased in untreated (saline) lesion (*p* < 0.001), WJ-MSCs (*p* < 0.001) and CM (*p* < 0.01) groups when compared to the healthy control levels. RANTES in CM and cM were significantly lower than the values in untreated lesion (*p* < 0.05 and *p* < 0.001), and also after the treatment with WJ-MSCs (*p* < 0.01 and *p* < 0.001). Moreover, the level of RANTES in the serum was significantly higher after CM than after cM treatment (*p* < 0.05). Significantly higher levels of MIP-1α and IL2 were found after WJ-MSC treatment when compared to the healthy control levels (both *p* < 0.05). TNFα and IL-1β enhancement after WJ-MSC treatment was significantly higher when compared to all other groups (all *p* < 0.01).

## 3. Discussion

MSC based therapy is a promising treatment strategy in SCI injury repair, however direct transplantation of MSCs to target tissues remains challenging due to poor cell engraftment and survival. Therefore, application of cell-free products based on MSC-derived secretome is considered as an advantageous alternative, reducing the immune compatibility problems, with effective mass-production and storage, and off-the-shelf availability.

We have previously demonstrated that intrathecally grafted human MSCs, although disappearing ~3 weeks after transplantation, enhanced locomotor recovery and axonal sprouting after SCI, and that this effect is potentiated by repeated application and gradually increases with the cell dose [8,36]. Based on these results, we investigated in this study whether CM derived from WJ-MSCs could have a similar regenerative effect as direct cell transplantation.

With proteomic analysis, in agreement with other works [31], we confirmed that WJ-MSCs release a high amount of HGF and other important neurotrophic factors, such as FGF2, BDNF or bNGF, and that WJ-MSC derived CM promote axon growth of DRG neurons. Our studies and others previously have also demonstrated that WJ-MSCs and their CM have immunomodulatory and neuroprotective activity (Petrenko et al., submitted) [11,12,15].

Notably, it has been proposed that MSC paracrine activity can be modulated by multiple aspects of their extracellular environment, such as soluble factors, pro-inflammatory cytokines, oxygen tension, 3D culture or matrix mediated physical and mechanical cues [13,37,38]. However, the precise mechanisms of MSC preconditioning by the microenvironment with respect to improving their regeneration/repair capacity, is still not elucidated. Since the aim of this study was to compare the effect of WJ-MSCs and their CM, we used unstimulated WJ-MSCs for cell transplantation and CM collection. Furthermore, to prevent serum starvation related cellular stress, which reduces cell proliferation and protein secretion into CM [39], and drives cells to apoptosis and superoxide production [40], we cultured WJ-MSCs in the medium supplemented with an insulin, transferrin, selenite supplement (ITS), similarly as was carried out in other reports using CM for the CNS treatment [41]. The combination of insulin, transferrin and selenite was optimized to prevent cell starvation and support growth and proliferation of cells in the serum-free medium.

To investigate the effect of WJ-MSCs and their CM in vivo, we used a clinically relevant balloon compression model of SCI. The WJ-MSCs were delivered in triple intrathecal application with a dose of 1.5 M, which we have demonstrated previously as the most effective dose in the SCI repair [8]. Correspondingly, CM derived from the same cell equivalent was collected and concentrated for the intrathecal delivery.

Behavioral evaluation confirmed a significant improvement of motor (BW test and BBB) and sensory (plantar test) functions after the treatment with WJ-MSCs, and CM as well as cM, as compared to the saline-treated rats. However, CM had a better recovery potential and revealed a significant improvement in BBB score 1 week, in the BW time 6 and 7 weeks and in the BW score 3, 7 and 8 weeks after SCI induction.

In addition to WJ-MSCs and CM, significant locomotor improvement was also found in the group that received control non-conditioned medium (cM) in comparison to the saline treated lesion. This effect was not surprising as cM contained ITS supplement. Insulin plays important roles in the maintenance of neural functions such as neuronal growth and differentiation, neuromodulation and neuroprotection [42]. Moreover, transferrin is an iron carrier and together with selenite, it may help to reduce oxidative stress. Interestingly, the ITS supplement has already been approved by the FDA as a medical device. Antioxidative and cell protective effects of these factors could therefore be considered as a complementary treatment for the SCI repair. However, despite its remarkable effect in behavioral tests, results of relative gene expression, in agreement with the results of axonal sprouting analysis, revealed the superior effect of the CM as well as WJ-MSCs over cM.

Histological analysis showed that lesion volume was significantly attenuated in both hWJ-MSC and CM treatment groups, which indicates that WJ-MSC secretome reduced secondary damage after SCI. However, only CM treatment further improved axonal sprouting and reduced the number of reactive astrocytes in the lesion area. These results are in line with the gene expression of markers related to neural growth and survival (*Gap43* and *Fgf2*), which were significantly enhanced in CM group, followed by the WJ-MSCs group, but not in the cM group.

However, the number of reactive astrocytes was reduced after CM and cM, but not after WJ-MSCs treatment, which corresponds with the significant down-regulation of *Gfap* expression in CM and cM groups, but not in the WJ-MSCs group, when compared to the saline control 9 weeks after lesion induction.

Remarkably, to reveal the effect of CM, the animals did not receive adjuvant immunosuppressive treatment, which enhances grafted cell survival and recovery following SCI [43]. Subsequently, without immunosuppression, WJ-MSCs treatment clearly increased the expression of inflammatory and chemotactic factors RANTES, IL2, IL1β, TNFα and MIP-1α in serum, and RANTES in CSF 4 weeks after the lesion induction, which was also confirmed by an increase in the relative expression of these genes in the same time interval. This finding indicates that despite the declared immune-privileged properties of MSCs, there was a systemic immune reaction against the xenogeneic cell transplantation, which, on the other hand, does not interfere with the regenerative effect of the WJ-MSCs on the SCI observed in locomotor and sensory functions, tissue sparing, axonal sprouting and gene expressions.

Of note, we did not perform in vivo analysis of cell survival within the vertebral canal, but due to the enhanced immune response, we expect a lower cell survival time as in our previous studies, where surviving cells were detected up to 2 weeks after the intrathecal MSCs transplantation under immunosuppression [8].

In contrast to the inflammatory response to WJ-MSC transplantation, CM treatment had an anti-inflammatory effect, confirmed by decreased gene expressions of *Ccl3*, *Ccl5*, *Tnfa*, *Irf5* and *Mrc1*, 4 weeks after SCI.

Several underlying mechanisms of neuroregenerative action of MSC-derived secretome have been suggested in SCI repair. Lu et al. [26] proposed that systemic administration of extracellular vesicles after SCI reduced pericyte migration via downregulation of NF-kB p65 signaling, with a consequent decrease in the permeability of the blood-spinal cord barrier. Tsai et al. [24] found that systemic administration of CM derived from BM-MSCs upregulated the protein levels of Olig 2 and HSP70 and increased autophage-related proteins in the injured spinal cords. Lankford et al. [44] demonstrated that intravenously delivered exosomes derived from MSCs were taken up by M2 macrophages within the SCI lesion and spleen and suggests the idea of systemic immune responses to exosome infusion in SCI recovery.

Of note, various neurotrophic factors that were identified in the WJ-MSCs derived CM, such as HGF, FGF2 and BDNF, have been reported to be effective for the SCI repair [45,46]. In our study, we proposed that the improvement of histo-morphological parameters together with sensory and motor functions result from an activation of endogenous neuro-restorative processes orchestrated by neuromodulatory factors secreted by WJ-MSCs.

In conclusion, we have shown that repeated intrathecal delivery of CM derived from human WJ-MSCs enhanced tissue preservation and axonal growth into the lesion, reduced inflammation and glial scaring and improved functional recovery in a clinically relevant model of acute SCI. We confirmed that CM represents a promising alternative to direct stem cell transplantation.

## 4. Materials and Methods

### 4.1. Isolation and Culture of WJ-MSCs

Discarded human umbilical cords (UC) were obtained from healthy full-term neonates (*n* = 3) after spontaneous delivery at University Hospital (Pilsen, Czech Republic), with the informed consent of the donors. All the studies using human UC or WJ-MSCs were approved by the local ethics committee. Human umbilical cords were donated anonymously. About 10 cm of UC was aseptically stored in sterile phosphate buffered saline (PBS; IKEM, Prague, Czech Republic) with antibiotic–antimycotic solution (AA; Sigma-Aldrich, St Louis, MO, USA) at 4 °C and transported to the laboratory within 24 h. After washing several times in PBS-AA, UC were rinsed in 10% Betadine (EGIS Pharmaceuticals PLC, Budapest, Hungary), blood vessels were removed and the remaining Wharton’s jelly tissue was chopped into small fragments (1–2 mm^3^). WJ-MSCs were isolated by digestion fragments in 0.26 U/mL Liberase^TM^ (Roche Custom Biotech, Mannheim, Germany) [47] and 1 mg/mL hyaluronidase (Sigma-Aldrich, St. Louis, MO, USA) in PBS-AA solution at 37 °C with constant shaking for 2 h. After centrifugation (450× *g* 10 min) cells and undigested fragments were diluted in a complete culture medium (CCM) containing the alpha-minimum essential medium (αMEM; LONZA, Basel, Switzerland), 5% pooled platelet lysate (PL; Bioinova, Ltd., Prague, Czech Republic) and 10 µg/mL gentamicin (Sandoz, Holzkirchen, Germany), repeatedly pipetted with a Pasteur pipette to mechanically liberate individual cells, and filtered through a 40-µm cell strainer (Falcon, Thermo Fisher Scientific, Waltham, MA, USA). Isolated cells were cultured at 37 °C, in a humidified atmosphere, containing 5% CO_2_ with regular media changes twice a week. Pooled WJ-MSCs at passage 3 were used for cell transplantation and CM production.

In respect of possible future clinical applications, human WJ-MSCs and CM were produced in xeno-free culture conditions for all of our experiments.

### 4.2. Production of Conditioned Medium (CM)

To prepare conditioned medium, WJ-MSCs at passage 3 pooled from three donors were seeded at a density of 5 × 10^3^ cells/cm^2^ into T-175 Nunc™ culture flasks (NUNC, Thermo Fisher Scientific, Waltham, MA, USA), and cultured in CCM till 80%–90% confluence. Then, the CCM was removed and after three washes with PBS replaced by 10 mL PL-free αMEM supplemented with 1×x Insulin-Transferrin-Selenium (ITS; Thermo Fisher Scientific, Waltham, MA, USA). After 24 h, WJ-MSC derived CM was collected, centrifuged at 1500 rpm for 10 min and filtered through a 0.22 µm filter (TPP, Trasadingen, Switzerland). Concurrently, WJ-MSCs were harvested using 0.05% Trypsin/EDTA (Thermo Fisher Scientific) and the number of cells producing CM were calculated using Burker hemocytometer (P-Lab, Prague, Czech Republic). CM was concentrated (20×) using 3 kDa cut-off Amicon Ultra Centrifugal Filters (Sigma-Aldrich, St. Louis, MO, USA) to obtain a concentration equivalent to 1.5 M cells/50 µL. Concentrated samples were stored at −80 °C until use. Three aliquots of unconcentrated CM were stored for proteomic analysis. PL-free αMEM supplemented with ITS cultivated for 24 h at 37 °C in 5% CO_2_, 20× concentrated and stored at −80 °C was used as the non-conditioned control medium (cM).

### 4.3. Proteomic Analysis of CM

Concentrations of cytokines and growth factors including BDNF, SDF-1α, VEGF-A, sICAM-1, sVCAM-1, FGF-2, HGF, MCP-1, IL-6 and NGF were assessed by Luminex^®^-based multiplex ProcartaPlex^®^ Immunoassay (eBioscience, Affymetrix, Bender MedSystems GMBH, Vienna, Austria, now Thermo Fisher Scientific, Waltham, MA, USA). The samples were analyzed in duplicates.

### 4.4. Balloon Compression Spinal Cord Injury

70 male Wistar rats (250–300 g, Velaz, Czech Republic) were used for the experiments. All experiments were performed in accordance with the European Communities Council Directive of 22nd of September 2010 (2010/63/EU) regarding the use of animals in research and were approved by the Ethics Committee of the Institute of Experimental Medicine, Academy of Sciences of the Czech Republic in Prague.

As a model of SCI, a balloon-induced ischemic-compression lesion was used [48]. At the beginning of the surgery the animals were anesthetized with Isoflurane (Forane; Abbott Laboratories, Queenborough, UK), analgesia was induced by subcutaneous injection of buprenorphine (0.05–0.1 mg/kg, Orion Pharma, Prague, Czech Republic) and surgical prophylaxis was maintained by intramuscular injection gentamicin sulphate (Lek Pharmaceutical 5 mg/kg). After the skin incision, the paravertebral muscles were separated at the level of thoracic vertebra T7–T12 and laminectomy of T 10 was performed. A sterile 2-french Fogarty catheter was carefully inserted into the epidural space until the centre of the balloon rested on the level of thoracic vertebra 8 (T8). The balloon was rapidly inflated with 15 μL saline and kept for 5 min. During this procedure, 3.5% isoflurane in air was administered at a flow rate of 0.3 L/min, and the animal’s body temperature was kept at 37 °C with a heating pad. After 5 min the catheter was rapidly deflated and removed, and separated muscles and incised skin were sutured by single non-absorbable stitches. The lesioned animals were assisted in feeding and urination until they had recovered sufficiently to perform these functions on their own. The animals received gentamicin sulphate (5 mg/kg) for 7 days to prevent postoperative infections and were allowed to feed and drink ad libitum.

### 4.5. Transplantation Procedure

Transplantation of hWJ-MSCs or application of CM was performed on the 1st, 2nd and 3rd week after the SCI. Treatment was given intrathecally by a lumbar puncture between L4 and L5 through a 25 G needle under the short-time general anesthesia described above. After injection the needle was rested in situ for 30 s to prevent backflow of the content. Animals were divided into four groups. The first group received 1.5 M of hWJ-MSCs in 50 μL PBS (*n* = 18). The second group received a CM in a concentration equivalent of 1.5 M cells in 50 μL (*n* = 15). The third group received 50 μL control non-conditioned medium cM (*n* = 17). The fourth (saline) control group received an injection of 50 μL saline (*n* = 20). Animals (*n* = 49) surviving 9 weeks were used for behavioral testing, histological, immunohistochemical and qPCR analysis. For evaluation of qPCR and cytokine levels in plasma and CSF, some rats (*n* = 21) were sacrificed 4 weeks after SCI and these animals were not included in behavioral testing.

### 4.6. Behavioral Testing

Locomotor and sensory functional tests were performed before SCI and then every week for the 8 weeks starting the first week after SCI in the case of BBB and plantar test, and every week starting with the 3rd week after SCI in the case of the Beam walk test.

#### 4.6.1. BBB Test

The BBB open field test was used to evaluate basic locomotor functions: The joint movement, weight support, forelimb–hindlimb coordination, paw placement and stability [49]. The rats were placed on the floor surrounded by paper boundaries, defining a space of rectangular shape, for approximately 4 min. Results were evaluated in the range of 0–21 points: 0 indicated complete motor incapability and 21 indicated a healthy state. Each hindlimb was evaluated separately. The BBB score was calculated as a mean value from the scores of both legs.

#### 4.6.2. Beam Walk Test

Beam walk test assesses a rat’s motor function and limb coordination by testing its ability to cross a 1 m long narrow (3.4 cm) beam with a flat surface. The latency and the trajectory to traverse the beam were recorded by a video tracking system (TSE-Systems Inc., Bad Homburg, Germany) for a maximum of 60 s. Performance of locomotor coordination was evaluated using a 0–7 point scale modified from [50].

#### 4.6.3. Plantar Test

A standard Ugo Basile test apparatus (Ugo Basile, Comerio, Gemonio, Italy) was used for the plantar (hot plate) test. The rats were placed in separate semi-transparent acrylic boxes, which protected them from watching each other, and a mobile device with an infrared heating lamp was positioned below the targeted hindlimb paw, always within the same region. Latency in withdrawal response to thermal nociceptive stimulus was automatically measured. Each paw was stimulated five times, with a stable pause-measurement interval. The latency of withdrawal of a healthy trained individual is usually from 8 to 12 s. The definition of hyperalgesia is a significant decrease of withdrawal latency.

### 4.7. Histological and Immunohistochemical Analyses

At the end of the experiment (9 weeks after the SCI), all animals were deeply anesthetized with ketamine (100 mg/kg) and xylazine (20 mg/kg) and transcardially perfused with PBS, followed by 4% paraformaldehyde in 0.1 M PBS. The spinal cord was dissected and removed from the spinal column and embedded in a paraffin wax. Serial cross-sections (5 μm thick) were obtained by a microtome within a 2 cm-long segment around the centre of the lesion. Each sample of the spinal cord was cut at 1 mm intervals. Samples were analyzed with an Axioskop 2 plus microscope (Zeiss, Oberkochen, Germany), ImageJ software (NIH) and TissueQuest analysis software (TissueGnostics, Vienna, Austria), seven sections cranially and caudally from the centre of lesion.

Total volume of spared white and gray matter of the spinal cord was distinguished using cresyl violet and luxol fast blue (both from Sigma) staining and analyzed by ImageJ software.

To analyze the extent of the glial scaring and number of reactive astrocytes, immunohistochemical analysis of a CY3-conjugated primary antibody against GFAP (Sigma) was used. The GFAP positive area around the central cavity together with the number of protoplasmic astrocytes was measured on five sections cranially and five sections caudally from the centre of lesion using ImageJ software.

In analysis of axonal sprouting, newly sprouted axons were visualized immunohistochemically using a primary antibody against GAP43 (Millipore, Billerica, MA, USA). Goat anti-mouse IgM Cy3 (1:200; Merck-Millipore, Germany) was used as the secondary antibody. Acquired images were visualized by TissueQuest software and the number of GAP43-positive fibers per section was manually counted on the five cross sections cranially from the centre of the lesion.

### 4.8. Gene Expression Analysis

Changes in the mRNA expression were determined by quantitative real-time PCR (qPCR) at 4 and 9 weeks after hWJ-MSCs administration (four animals from each group). The following rat genes were selected to make the multi-sided screening of different cell functions: Chemokines (*Ccl3* and *Ccl5*), genes associated with immunomodulation and inflammation (*Tnfα*, *IL-1β*, *Mrc1*, *Irf5* and *Nf-κb*), as well as genes related to neuroregeneration (*Fgf2*, *Gap43* and *Gfap*). Samples of the spinal cord around the centre of the lesion from rats sacrificed in the 4th week after SCI were stored in RNA later ^TM^ Solution (Invitrogen by Thermo Fisher Scientific, Waltham, MA, USA) at −80 °C and RNA was isolated using an RNeasy Lipid Tissue Mini Kit (Qiagen, Germany) according to the manufacturer’s instructions. In the case of the 9th week group, RNA was isolated from the paraffin cross sections of spinal cords around the centre of the lesion using the RNeasy^R^ FFPE Kit (Qiagen, Hilden, Germany) according to the manufacturer’s instructions. RNA amounts were quantified using NanoPhotometer^®^ P330 (Implen, München, Germany) and reverse transcribed into complementary DNA using the Transcriptor Universal cDNA Master (Roche, Basel, Switzerland) and T100TM Thermal Cycler (Bio-Rad, Hercules, CA, USA). The qPCR reactions were performed using cDNA solution, FastStart Universal Probe Master (Roche, Basel, Switzerland) and the following TaqMan^®^ Gene Expression Assays (Life Technologies by Thermo Fisher Scientific, Waltham, MA, USA): *Gapdh*/Rn01775763_m1, *Ccl3*/Rn01464736_g1, *Ccl5*/Rn00579590_m1, *Tnfa*/Rn01525859_g1, *IL1b*/Rn00580432_m1, *Mrc1*/Rn01487342_m1, *Irf5*/Rn01500522_m1, *Nfκb* /Rn01399572_m1, *Fgf2*/Rn00570809_m1, *Gap43*/Rn01474579_m1 and *Gfap* /Rn00566603_m1. The qPCR was carried out in a final volume of 10 µL containing cDNA equivalent of 65 ng of extracted RNA. Amplification was performed on the real-time PCR cycler (StepOnePlus^TM^, Life Technologies, Carlsbad, CA, USA). All amplifications were run under the same cycling conditions: 2 min at 50 °C, 10 min at 95 °C, followed by 40 cycles of 15 s at 95 °C and 1 min at 60 °C. All gained results were analyzed with StepOnePlus^®^ software version 2.3 with *Gapdh* as a reference gene (Carlsbad, CA, USA). The ΔΔ*C*t method was used for relative quantification of gene expression to saline treated animals.

### 4.9. Proteomic Analysis of CSF and Serum

The analysis of cerebrospinal fluid and serum was done 4 weeks after SCI. Before sacrificing of the rats for qPCR spinal cord tissue analysis; the CSF and blood was collected under general anesthesia with isoflurane (3.5 vol%, Forane, San Juan, PR, USA) combined with buprenorphine (0.05–0.1 mg/kg, Orion Pharma, Prague, Czech Republic). The blood was collected from the retroorbital sinus using a Pasteur pipette in a total volume of 2 mL. The blood was kept in vials without an anticoagulant at room temperature for up to 120 min to enable clotting. Then, the vials with blood were centrifuged two times at 2000× *g* for 5 min and the eluted serum was collected and immediately stored at −80 °C. The CSF was collected from the cisterna magna. The rat was flexed downward of the head at approximately 60 degree. A small incision was made in the skin, the muscles were separated and the atlanto-occipital membrane (the membrane between the occipital bone and the upper cervical vertebra) was exposed to be clearly visible. A 23 G needle connected to the 1 mL syringe was used to puncture the cisterna magna under the microscope and approximately 100–250 µL of cerebrospinal fluid was drawn from each rat.

Concentrations of cytokines and chemokines including interleukins IL-1β, IL-2, IL-4, IL-6, IL-10, IL-12p70, interferon gamma (IFN-γ), tumor necrosis factor alpha (TNF-α), macrophage inflammatory protein 1 alpha (MIP-1α, CCL3) and RANTES (CCL5) were assessed by ProcartaPlex^®^ Immunoassay (Thermo Fisher Scientific, Waltham, MA, USA) using rat Simplex™ and Basic Kits and Bio-Plex^®^ 200 System (eBioscience, Affymetrix, Bender MedSystems GMBH, Vienna, Austria, now Thermo Fisher Scientific, Waltham, MA, USA). The mean fluorescence intensities (MFI) were used for five parameter logistic regression standard curve fitting and quantitation of cytokine concentrations. The samples were analyzed in duplicates. Plasma obtained from untreated rats (*n* = 3) served as a negative (healthy) control.

### 4.10. Statistical Analysis

Data are presented as mean ± standard error of mean (SEM), statistical significance was analyzed using SigmaPlot V13 (Systat Software Inc., San Jose, CA, USA).

All behavioral tests were evaluated by two-way ANOVA RM with Student-Newman–Keuls post hoc test, *p* ≤ 0.05 *, *p* ≤ 0.01 ** and *p* ≤ 0.001 ***.

All histological analyses were evaluated by one-way ANOVA with Student-Newman–Keuls post hoc test, *p* ≤ 0.05 *, *p* ≤ 0.01 ** and *p* ≤ 0.001 ***.

All Δ*C*t values in gene expression studies were evaluated by one-way ANOVA with Student-Newman–Keuls post hoc test, *p* ≤ 0.05 *, *p* ≤ 0.01 ** and *p* ≤ 0.001 ***.

For immunomodulatory properties, a DRG-neurite outgrowth assay Mann–Whitney nonparametric test was used.

## Figures and Tables

**Figure 1 ijms-20-04516-f001:**
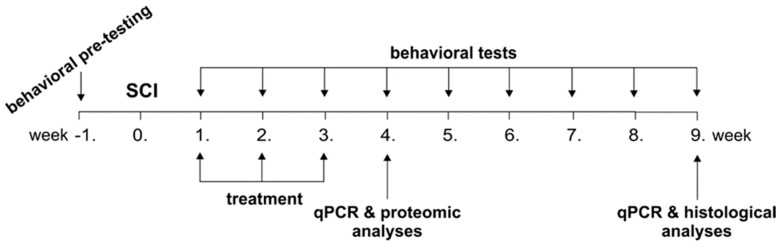
Time schedule of experiments. Following a spinal cord injury (SCI), the rats were treated by saline (ctrl), non-conditioned (control) medium (cM), conditioned medium (CM) and Wharton’s jelly derived mesenchymal stromal cells (WJ-MSCs) in the 1st, 2nd and 3rd week. The rats were behaviorally tested weekly up to the 9th week and analysed using qPCR (4th and 9th week), proteomic (4th week) and histological (9th week) analyses.

**Figure 2 ijms-20-04516-f002:**
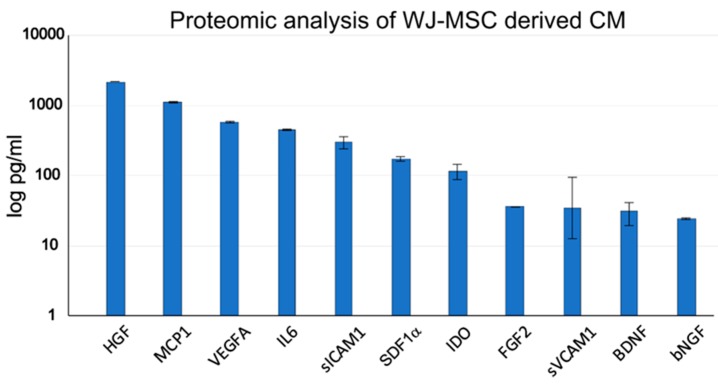
Proteomic analysis of the CM derived from WJ-MSCs.

**Figure 3 ijms-20-04516-f003:**
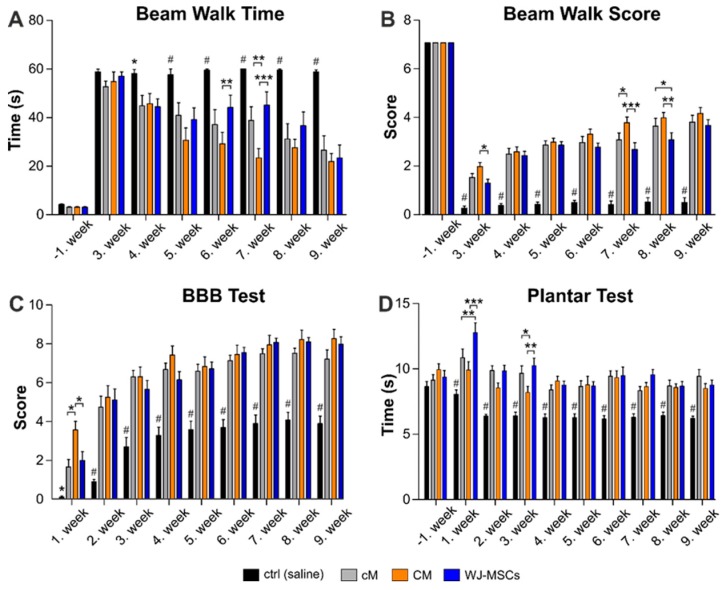
Results of behavioral testing. The locomotor and sensory function of animals were tested by a beam walk time measurement (**A**), beam walk score (**B**), Basso, Beattie and Bresnahan test (BBB) test (**C**) and plantar test (**D**). The number of animals in the groups is as follows: Saline treated control group *n* = 16, cM group *n* = 11, CM group *n* = 10 and WJ-MSCs group *n* = 12. Asterisks (*) and (#) above ctrl (saline) group columns show statistical significance of ctrl (saline) group vs. all particular treatments, which was * *p* < (0.05–0.01) and # *p* < 0.001. The significance between the treated groups: * *p* < 0.05, ** *p*< 0.01 and *** *p* < 0.001.

**Figure 4 ijms-20-04516-f004:**
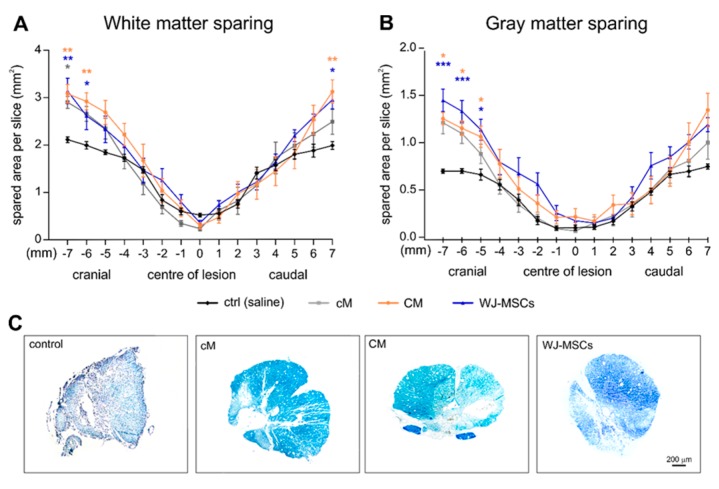
White matter (**A**) and gray matter (**B**) sparing expressed as spared area (mm^2^) per slice through the lesion area. Asterisks show the statistical significance of a particular treatment when compared to the saline treated control group. The number of animals in the groups is as follows: Saline treated control group *n* = 7, cM group *n* = 9, CM group *n* = 9 and WJ-MSCs group *n* = 9. (**C**) Representative images of spinal cord cross sections 5 mm cranially from the centre of lesion were stained using cresyl-violet luxol fast blue.

**Figure 5 ijms-20-04516-f005:**
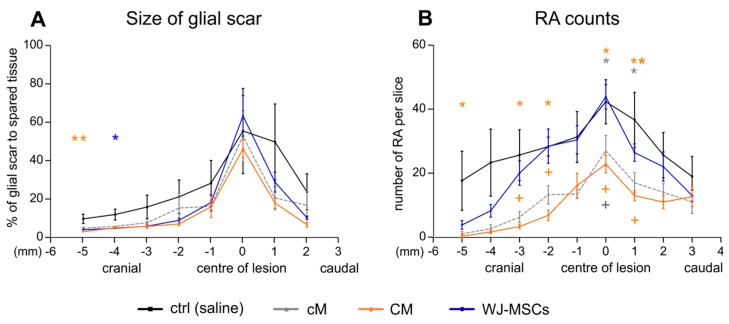
Glial scar formation analyses. (**A**) The size of glial scar expressed as the % to spared tissue. (**B**) Number of reactive astrocytes (RA) counted from the glial fibrillary acidic protein (GFAP) staining as shown in Figure 6. Number of animals: Saline treated control group *n* = 4, cM group *n* = 5, CM group *n* = 5 and WJ-MSCs group *n* = 5. Asterisks (*) show statistical significance of particular treatment, when compared to saline treated control group, crosses marks (+) show statistically significant difference between WJ-MSCs group and CM group (orange cross) and between WJ-MSCs group and cM group (gray cross), respectively. * *p* < 0.05, ** *p* < 0.01 and + *p* < 0.05.

**Figure 6 ijms-20-04516-f006:**
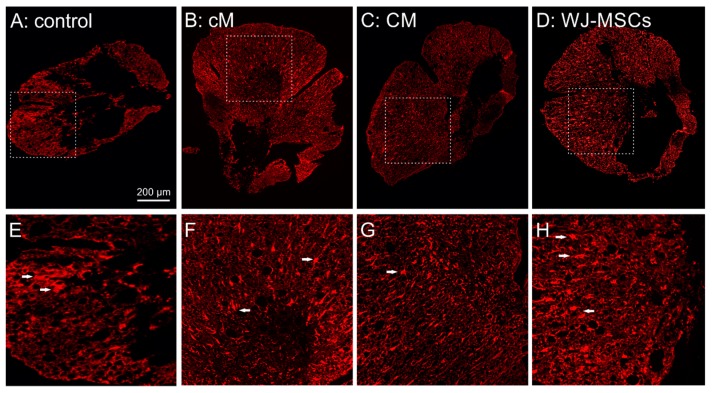
(**A**–**D**) Representative fluorescent images of GFAP staining of cross sections 9 weeks after the SCI for evaluation of the glial scar in the cranial part of the lesion, 5 mm from the centre. (**E**–**H**) highlight the details of the marked area in (**A**–**D**), with reactive astrocytes depicted by white arrows.

**Figure 7 ijms-20-04516-f007:**
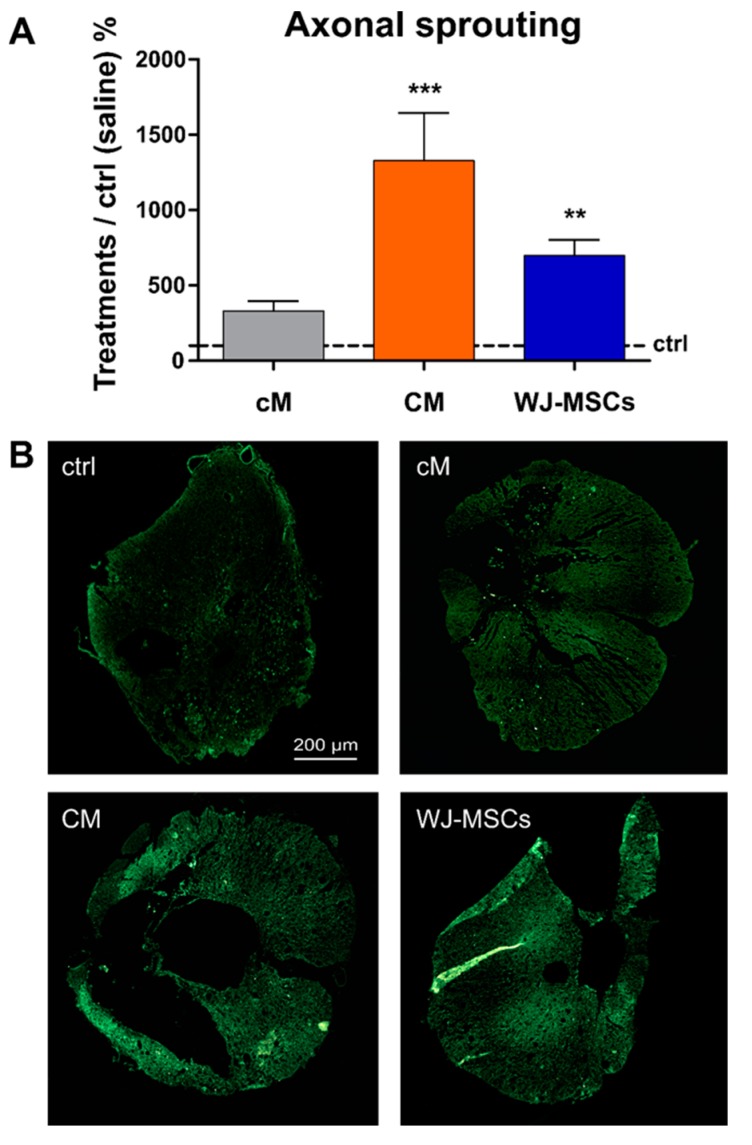
(**A**) The average number of GAP43+ fibers in the treated groups presented as relative when compared to the control (saline), which was set as 100% (depicted with a black interrupted line). Number of animals (five sections per animal): Saline treated control group *n* = 11, cM group *n* = 5, CM group *n* = 9 and WJ-MSCs group *n* = 7, ** *p* < 0.01 and *** *p* < 0.001. (**B**) Representative fluorescent images of GAP43 staining of cross sections 9 weeks after the SCI in the cranial part of the lesion, 5 mm from the centre.

**Figure 8 ijms-20-04516-f008:**
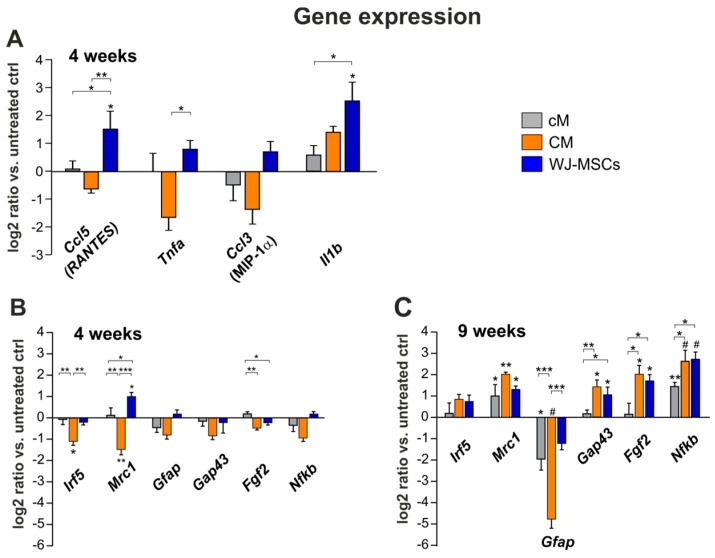
Relative expression of selected genes in the lesion 4 (**A**,**B**) and 9 weeks (**C**) after SCI. The graphs show the log2-fold changes of the ΔΔ*C*t values of the indicated genes in comparison to the animals treated with the saline. *n* = 4 per group, asterisks show statistical significance * *p* < 0.05, ** *p* < 0.01 and # *p* < 0.001 of a particular treatment, when compared to the saline treated control group. The significance between treated groups is * *p* < 0.05, ** *p* < 0.01 and *** *p* < 0.001.

**Figure 9 ijms-20-04516-f009:**
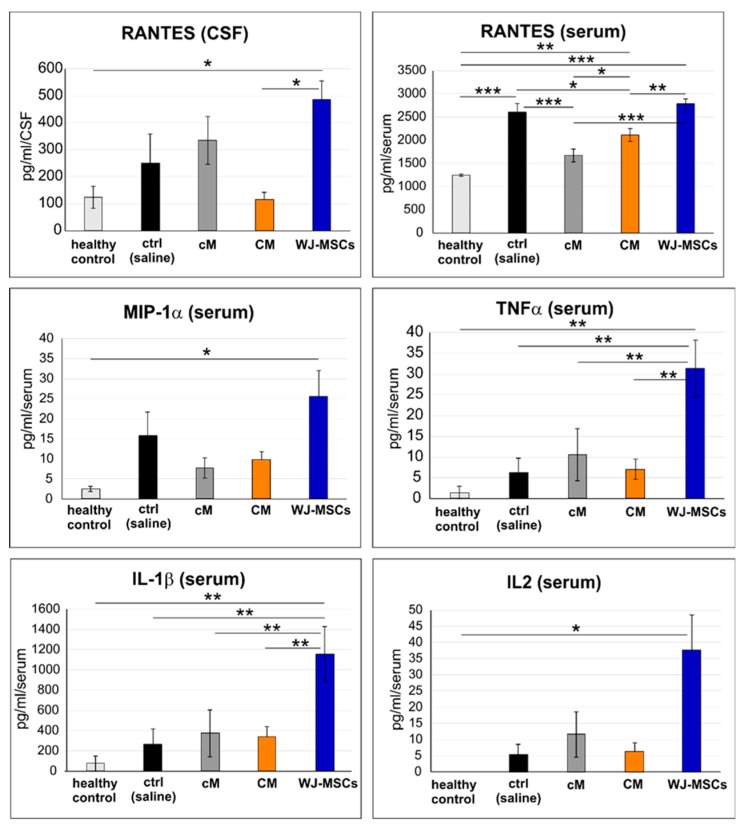
Protein levels in cerebrospinal fluid (CSF) and blood serum taken 4 weeks after SCI. The number of animals: *n* = 3 for healthy control, *n* = 4 for control (saline), *n* = 4 for cM, *n* = 5 for CM and *n* = 5 for WJ-MSCs. * *p* < 0.05, ** *p* < 0.01 and *** *p* < 0.001.

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
