# Peer review of "The Effect of Wharton Jelly-Derived Mesenchymal Stromal Cells and Their Conditioned Media in the Treatment of a Rat Spinal Cord Injury"

_ijms, 2019, doi:10.3390/ijms20184516_

Round 1

Reviewer 1 Report

This is an interesting study on comparing the therapeutic effects on SCI between transplantation of Wharton Jelly-Derived Mesenchymal Stromal Cells and their condition medium. Overall, the research provides important information on the application of stem cells derived condition medium in SCI, including the proteomic analysis and the gene expression. There are some minor suggestions and concerns:

1. The Wharton Jelly-Derived Mesenchymal Stromal Cells need to be characterized before transplantation and collection of condition medium.

2. It would be better to show the representative images of GAP43+ fibers in different groups.

3. The figure A1 in the supplementary figure, the y-axis is the % area, how did you quantify to get the % area? Why you choose %area instead of axon length? The method for quantification of the % area is not clear.

Author Response

Comment 1: The Wharton Jelly-Derived Mesenchymal Stromal Cells need to be characterized before transplantation and collection of condition medium.

Response 1: WJ-MSCs, that were used for the production of conditioned media and for the transplantation, were analysed for surface marker expression and multipotent differentiation potential in the 3rd passage as is described in our previous article written by Krupa, P.; Vackova, I.; Ruzicka, J.; Zaviskova, K.; Dubisova, J.; Koci, Z.; Turnovcova, K.; Urdzikova, L. M.; Kubinova, S.; Rehak, S.; Jendelova, P., The Effect of Human Mesenchymal Stem Cells Derived from Wharton's Jelly in Spinal Cord Injury Treatment Is Dose-Dependent and Can Be Facilitated by Repeated Application. Int J Mol Sci 2018, 19, (5).

All hWJ-MSC batches fulfilled the ISCT criteria; they were plastic adherent, positive for CD105, CD73 and CD90 and lacked expression of CD45, CD34, CD14 and HLA-DR surface molecules, and differentiated to osteoblasts, adipocytes and chondroblasts in vitro.

We added information about MSC characterisation into the manuscript page 4, line 121.

Comment 2: It would be better to show the representative images of GAP43+ fibers in different groups.

Response 2: We added representative images of GAP43 staining to Figure 7.

Comment 3: The figure A1 in the supplementary figure, the y-axis is the % area, how did you quantify to get the % area? Why you choose %area instead of axon length? The method for quantification of the % area is not clear.

Response 3: To quantify the axon growth from DRG, we analysed the area (%) of beta III tubulin positive staining using Image J software. The axon length was not possible analyse, due to the high axonal density in the culture. To analyse axon length, we would need the lower DRG density within the culture. However, as it was apparent that CM stimulates axon growth, we did not repeat the experiments with lower DRG density to reduce the number of animals.

We modified the text in the Figure 1A to make the analysis more clear.

Reviewer 2 Report

The present manuscript “The effect of Wharton Jelly-Derived Mesenchymal Stromal Cells and their conditioned media in the treatment of rat spinal cord injury”, by Chudickova M et.al. Previously they have established the protective effect of Wharton’s Jelly-Derived Mesenchymal Stromal Cells (WJ-MSCs) in spinal cord injury (SCI). In the present manuscript hey have demonstrated the potential use of conditioned media (CM) derived from WJ-MSCs in treating SCI.

They have shown that both WJ-MSCs and CM treatment resulted in a higher amount of spared gray and white matter and enhanced axonal growth, however CM only improved axonal sprouting and reduced the number of reactive astrocytes in the lesion area.

Author Response

We thank to the reviewer for the evaluation.